# Using Schlieren Imaging and a Radar Acoustic Sounding System for the Detection of Close-in Air Turbulence [note 1]

**DOI:** 10.3390/s23198255

**Published:** 2023-10-05

**Authors:** Samantha Gordon, Graham Brooker

**Affiliations:** ACFR/AMME, University of Sydney, Darlington 2008, Australia; sgor5338@uni.sydney.edu.au

**Keywords:** ultrasound-modulated Schlieren imaging, radio acoustic sounding, RASS, imaging, ultrasound

## Abstract

This paper presents a novel sensor for the detection and characterization of regions of air turbulence. As part of the ground truth process, it consists of a combined Schlieren imager and a Radar Acoustic Sounding System (RASS) to produce dual-modality “images” of air movement within the measurement volume. The ultrasound-modulated Schlieren imager consists of a strobed point light source, parabolic mirror, light block, and camera, which are controlled by two laptops. It provides a fine-scale projection of the acoustic pulse-modulated air turbulence through the measurement volume. The narrow beam 40 kHz/17 GHz RASS produces spectra based on Bragg-enhanced Doppler radar reflections from the acoustic pulse as it travels. Tests using artificially generated air vortices showed some disruption of the Schlieren image and of the RASS spectrogram. This should allow the higher-resolution Schlieren images to identify the turbulence mechanisms that are disrupting the RASS spectra. The objective of this combined sensor is to have the Schlieren component inform the interpretation of RASS spectra to allow the latter to be used as a stand-alone sensor on a UAV.

## 1. Introduction

Turbulence is a well-known problem for aircraft as it can cause them to behave unpredictably and become difficult to control, with potentially catastrophic consequences [1,2,3]. The impact of turbulence is most pronounced for small aircraft, most typically unmanned aerial vehicles (UAVs) [4]. According to Gao et al. [5], the American National Standards Institute (ANSI) has identified weather robustness as a high-priority research area. Unfortunately, to date, few publications have addressed the weather effects on UAV performance and safety. Atmospheric phenomena like air temperature, wind speed and precipitation have been shown to affect UAV endurance, control, aerodynamics, airframe integrity, airspace monitoring as well as sensors for collision avoidance and navigation. But understanding where and how these conditions arise and their impact on UAV operations is complicated. Minimizing the impact of turbulence through pre-emptive control can improve the smoothness, speed and safety of a flight and allow UAVs to operate under conditions that would have previously been unsafe or even impossible [6,7].

There have been several attempts to predict turbulence using meteorological information (e.g., [8,9,10]). Unfortunately, current predictive models do not work well for all types of turbulence [11]. In 2017, the Japan Aerospace Exploration Agency (JAXA) developed a lidar-based sensor that could be attached to aircraft to detect clear air turbulence (CAT) in their path, and this has been effective in warning pilots, particularly on landing approach, that instability can be expected and additional vigilance is required [12]. There is potential to use a radio acoustic sounding system (RASS) to perform a similar function that can be implemented on much smaller aircraft including UAVs.

The scattering of light at acoustically induced variations in permittivity has been known since 1932. However, the first radar using the reflection of sound waves was demonstrated for the first time in 1961 [13].

In an attempt to visualize air turbulence with the best possible resolution and to quantify its effects on the RASS, we present a dual-modality sensor that integrates Schlieren imaging and RASS to provide a proof-of-concept of this capability. Both Schlieren imaging and RASS have been used independently but we understand they have never been combined to detect turbulence. The objective of this research is to have the Schlieren component inform the interpretation of the RASS spectra so that it could ultimately be used as a stand-alone sensor on a UAV.

The first Schlieren imagers are believed to have been independently created by Robert Hooke and Christiaan Huygens in the 1670s [14,15]. Although technology has advanced significantly since then, the same principles are still used for Schlieren imagers. They visualize refractive index gradients without interfering with the medium being probed [16].

The reflection of electromagnetic radiation from abrupt changes in atmospheric characteristics is now a well-known effect. From the beginnings of radar use in WWII, it was one of the phenomena that produced artifacts called “angels” [17]. However, it was not until the late 1950s that the changes in refractive index in air induced by acoustic signals were first identified [18]. Over the next 50 years, the phenomenon was used to produce progressively more sophisticated RASSs to examine air temperature, wind profiles and turbulence in the lower troposphere [19,20,21].

By the 1990s, RASS was being applied to indoor problems [22,23,24,25], and the application was being widened to other fields, such as Identification, Friend or Foe (IFF) [25] and the detection of aircraft wake vortices [26,27].

## 2. Operational Principles

As an acoustic wave propagates through the air, the density of the medium in one region periodically increases and decreases, making these peaks and troughs appear to travel in the direction of propagation. These changes in density result in subtle differences in the refractive index, *n*, between the peak and trough.

### 2.1. Schlieren Operational Principles

Schlieren imaging relies on Snell’s law to visualize refractive index gradients. Electromagnetic (EM) waves undergo an angular deflection along an axis perpendicular to the direction of travel ($\epsilon_{x}$ or $\epsilon_{y}$) when they are exposed to a change in refractive index n. The angular deflection is
(1)$$\epsilon_{y}=\frac{1}{n_{ref}}\int \frac{\partial n}{\partial y}dz=\frac{Z}{n_{ref}}\frac{\partial n}{\partial y}.$$
where $n_{ref}$ is the refractive index of the surrounding medium and z is the direction of propagation. Schlieren imagers can be divided into two categories: qualitative systems and quantitative systems.

Figure 1 shows a typical qualitative Schlieren imager. A point source produces light rays that are reflected off a spherical mirror, though a parabolic mirror will also suffice as the differences are negligible for large radii of curvature [14]. The light rays are refracted by refractive index gradients in front of the mirror as shown. A light block such as a sharp edge or wire is used to improve contrast before the reflected light rays are captured with a camera. While there are a range of ways to configure a Schlieren imager, they all rely on the same principles [16,28,29].

A light block such as the sharp edge shown in Figure 1 enhances the contrast in the image. As illustrated in Figure 2, some of the light rays that would pass by the light block if not refracted are instead blocked. The reverse also occurs where some rays that would have been blocked if they had not been refracted instead pass unaffected. The combination of light rays getting blocked or passing through when the opposite occurs without refraction produces light and dark regions, representing the changes in refractive index.

Unfortunately, such a light block places a fundamental limit on the system as only deflections perpendicular to the edge of the light block can be observed and so only refractive index gradients perpendicular to the edge of the light block can be analyzed [16]. The light block can be angled strategically, or multiple light blocks oriented at different angles can be used to show refractive index gradients along two dimensions, as noted by [30].

Unlike qualitative Schlieren imagers, quantitative systems can measure the magnitude of the deflection and provide more information about the refractive index gradients [16]. Background-oriented Schlieren (BOS) is a type of quantitative Schlieren where refractive index gradients are imaged in front of a known patterned background [31,32]. When imaged, the refractive index gradients make the background pattern appear distorted, and these distortions can be analyzed to quantify the deflection produced by refractive index gradients [28,33]. Another common type of quantitative Schlieren is rainbow Schlieren, where a rainbow gradient is used instead of a light block and the color imaged gives an indication of the magnitude of the deflection [34,35,36].

Qualitative systems are relatively simple to construct, and they provide an understanding of any turbulence. However, qualitative systems do not offer precise numerical information. Quantitative Schlieren systems allow for a more in-depth analysis than qualitative systems, but they are generally more complex to construct.

#### Schlieren Imaging and Acoustic Waves

The refractive index gradients comprising an acoustic wave can also be visualized with a Schlieren imager. Ref. [37] derives the relationship between the deflection of an electromagnetic wave, the sound pressure level (SPL), and the frequency of an acoustic wave. This relationship is plotted in Figure 3.

As noted by several papers (e.g., [37,38]), for a reasonable custom Schlieren imaging system, any acoustic wave at audible frequencies would need to be at a volume that is painful for humans to be clearly imaged. However, when considering ultrasonic frequencies, the amplitude of the sound wave does not need to be as large, as reflected in Figure 3. Also, ultrasonic frequencies are not within the human hearing range so are not painful to us though they might cause hearing damage with prolonged use.

The fast propagation speed of acoustic waves presents another challenge for visualizing them with Schlieren imaging. A short exposure time (fast shutter speed) is usually required for capturing fast-moving subjects. However, given the periodic nature of acoustic waves, a light source can be strobed at the same frequency as the acoustic wave to make the acoustic wave appear as if it is a standing wave [39]. Several images can then be integrated to improve the signal-to-noise ratio (SNR) of the resulting image [40]. While this is useful, the apparatus must not be adjusted in between images to accurately complete the integration.

From papers such as [37,41,42,43], it is evident that imaging audible acoustic waves with a Schlieren imaging system is difficult due the explained challenges. However, imaging ultrasonic signals in air is feasible and has been achieved in numerous situations, such as by [39], who examined 5 MHz ultrasonic waves using a pulsed laser. They did not use a light block but rather subtracted the image with the acoustic wave from an image without. There is also literature (e.g., [38,40,44]) that outlines Schlieren systems to image ultrasonic waves for educational purposes. While the outlined systems are all similar, they each present variations. They all use a qualitative Schlieren setup, as outlined above, with a strobed light-emitting diode (LED) to image ultrasonic acoustic waves. When the LED is strobed at the same frequency as the ultrasonic wave, the wave appears stationary. If the strobing frequency is slightly lower than the acoustic frequency, the acoustic waves appear to be moving away from the source, and if the strobing frequency is slightly higher, the motion appears to occur in the opposite direction, as described by sampling theory [45]. Genuine standing waves can be created if the sound waves are reflected off a flat mirror in the path of the signal [38]. If genuine standing waves are desired, a Schlieren system must be designed such that they can be produced and clearly imaged. Moreover, if genuine standing waves are not wanted, then the system must be designed to ensure they are not produced.

### 2.2. RASS Operational Principles

When an electromagnetic wave passes through the air, in which there are changes in density resulting from the propagation of an acoustic wave, a small fraction of the signal will be reflected at each of the density transitions.

#### 2.2.1. Bragg Matching

Bragg matching occurs where the electromagnetic wavelength is equal to twice the acoustic wavelength, which results in the tiny, reflected components adding in phase to form a larger return, as seen in Figure 4 for a 40 kHz acoustic signal.

The amplitude of the coherent sum increases linearly with the number of acoustic cycles, *N*. As the received echo power is proportional to the square of the amplitude, it will be proportional to *N*^2^. In the RASS case, the Bragg reflector is not static, but there is a pulse of sound travelling out from the transducer at a velocity *v*_a_ ≈ 340 m/s. It is easy to show that if the acoustic and electromagnetic sensors are collocated, a Doppler shift equal to the acoustic frequency, *f*_a_, occurs.
(2)$$f_{d}=\frac{v_{a}}{\lambda_{a}}=f_{a}.$$

#### 2.2.2. Focus Effect

One feature of collocating the acoustic and radar sensors is the focus effect shown in Figure 5. In this geometry, both the wave-fronts expand with the same radius of curvature and so coherence is maintained over the full area of the expanding pulse [25].

The radar cross-section (RCS), *σ*_a_ (m^2^), of this expanding acoustic pulse can be determined in terms of the acoustic power, *P*_a_, the acoustic antenna beamwidth, *θ*_a_ (rad), and the range, *R* (m). If the radar beam is wider than the acoustic beam [25], then
(3)$$\sigma_{a}=\frac{1.76\times 10^{-15}\times 4\pi^{5}R^{2}N^{2}P_{a}g_{a}{\left(1-\cos\frac{\theta_{a}}{2}\right)}^{2}}{16\lambda_{a}^{2}}.$$

With some simplification, (3) reduces to
(4)$$\sigma_{a}=1.69\times 10^{-12}R^{2}\theta_{a}^{2}N^{2}P_{a}$$
where *θ*_a_ (rad) is the acoustic beamwidth.

#### 2.2.3. Effect of Turbulence

The effects of turbulence are twofold. Firstly, local changes in the direction of the airflow can displace the acoustic pulse to reduce the effectiveness of the Bragg matching. Secondly, more global turbulence can affect the curvature of the pulse to reduce the focus effect for a collocated sensor configuration.

Together, these effects will reduce the effective RCS, with the result that the tracked pulse will be extinguished over a shorter distance than it would in still air. The rate at which the reduction in the echo return occurs is indicative of the magnitude of the turbulence in that direction. Additionally, temporal perturbations in the propagation speed based on turbulence can be manifest as micro-Doppler components of the base spectral line.

#### 2.2.4. Atmospheric Attenuation

The Equations (3) and (4) describing the RCS does not consider the attenuation of the acoustic signal, which increases significantly with increasing frequency [46,47]. At 40 kHz, the attenuation varies between 1.1 dB/m and 1.4 dB/m depending on the relative humidity. The attenuation corresponding to the range of operation should be subtracted from the RCS calculated in (4) to produce the effective value.

## 3. Materials and Methods

An RASS previously used in [48] was provided, while a custom Schlieren imager was constructed and iteratively optimized. A combined system was designed such that the RASS could be used in synchronization with Schlieren imaging. A schematic of this combined physical system is seen in Figure 6, and the connections between components are detailed in Figure A1.

### 3.1. Monostatic RASS Radar System

#### 3.1.1. System Configuration

The wavelength of an acoustic system operating at 40 kHz is 8.5 mm. Therefore, the radar system must operate at a wavelength of 17 mm to satisfy the Bragg condition. This equates to a frequency of 17.65 GHz. A conventional Doppler radar system with a reflected power canceller (RPC) was constructed from discrete components, as shown schematically in Figure 7.

The RASS performance was determined in simulation. The RCS defined in (4) and modified by the atmospheric attenuation is plotted in Figure 8. For an acoustic power of 1 W, it can be seen that the RCS reaches a maximum at a range of 6.5 m before falling off as the atmospheric attenuation begins to dominate over the *R*^2^ term.

The Doppler radar model assumes that the received signal-to-noise ratio is limited by thermal noise because the RPC cancels the phase noise-leakage effects. The system performance shown in Figure 9 is determined for the following parameters:Operational frequency 17.65 GHz;RF transmit power 24 dBm;Antenna gain 25 dB;Receive filter bandwidth 3 kHz;System noise figure 5 dB;100 pulses integrated.

Because the acquisition of the Doppler signal can be synchronized with the generation of an acoustic pulse, it is possible to integrate a large number of measurements to improve the overall SNR. For example, for a 10 m maximum range, each measurement takes 30 ms, so the coherent integration of 25 returns would only require 750 ms to perform.

The number of cycles in a pulse is selected depending on the spatial resolution required and the available SNR. For an acoustic wavelength of 8.5 mm, and N = 60, the pulse spans a range of 510 mm, which defines the spatial resolution for the received Doppler measurement. As the required range is decreased, the available SNR increases and fewer cycles need to be used, with a resulting improvement in the spatial resolution.

#### 3.1.2. Radar Receiver Requirements

Figure 9 predicts that the received Doppler echo power from an acoustic pulse will be something between −135 and −155 dBm at a range of 4 m. This signal needs to be amplified by at least 100 dB to reach mV levels suitable for the ADC board. This gain is achieved by a pair of cascaded RF amplifiers providing 53 dB of gain and an audio amplifier and filter with a further 57 dB of gain at 40 kHz, as shown in Figure 10.

The audio amplifier gain characteristics were configured with sufficient bandwidth to accommodate variations in the Doppler frequency of the received signal due to either changes in temperature, air movement, or the speed of the radar.

#### 3.1.3. System Hardware

The RASS was built using a polarized wire reflector to combine the acoustic and radar signals into a single beam, as documented by Weiβ [24] and shown in Figure 11. To minimize microphonics, which plagued earlier configurations of the system, the acoustic array was hung from springs (not visible). Connections from the RASS to other components are displayed in Figure A3.

#### 3.1.4. RASS System Calibration

To measure the system performance, it is convenient to use a Doppler reflector with a known RCS. Conventional moving targets are not suitable as the receiver is tuned to provide maximum sensitivity at around 40 kHz, which corresponds to a velocity of 340 m/s. In addition, the expected return is incredibly small, as can be seen in Figure 9.

To achieve this, a small Doppler target was developed using a 40 kHz piezo transducer and a small ball bearing. Sinusoidal excitation voltages of between 1 V and 10 V produced variations in the measured RCS from −140.5 dBm^2^ to −120.5 dBm^2^ [49]. Based on this known RCS and the measured voltage output by the radar, we were able to confirm the accuracy of our radar model to within about 1 dB using a process known as “closing the calibration”.

#### 3.1.5. Calibration of Turbulence Generation Methods

A number of techniques for generating turbulence were developed. These included two methods of generating spiral wind patterns: one based on a cooling fan and another on a powerful leaf blower. In both cases, the moving air was passed through the static blades of a fan to generate spiral turbulence. In addition, a “toy” vortex canon that generated torus-shaped rolling vortices similar to smoke rings was used. The turbulence was visualized by introducing smoke from a smoke generator, but no attempt was made to characterize it any further.

### 3.2. Ultrasound-Modulated Schlieren Imager

Our ultrasound-modulated Schlieren imager was similar to several previous systems that have observed acoustic waves at ultrasonic frequencies, including [38,39,40,44].

There were two main sections to the Schlieren imager, the first consisting of the mirror and its mounting and the second consisting of the camera, light source, light block and corresponding mounting system. Figure A2 gives an overview of the various components and connections comprising the Schlieren imager. A 25 cm diameter parabolic mirror removed from a telescope was mounted onto a wooden frame and placed at one end of a bench, while the camera and light ensemble was positioned approximately 2.5 m away (double the estimated focal length of the mirror).

A Blackfly BFLY-PGE-23S2C camera (Blackfly S camera) [50] with a 75 mm lens was interfaced with a custom Python script. The camera was mounted onto a wooden block with a metal elbow joint. The software used a software development kit (SDK) released by the camera manufacturer version 2.7.0.128 [51]. The software was divided into two parts. One part was run on an ASUS Zenbook UX303LN laptop (ASUS laptop) responsible for triggering the camera through an Analog Discovery 2 board [52]. The other part was run on a Microsoft Surface Pro 7 (Microsoft Surface) to configure the camera settings as well as read and save the captured images. The Microsoft Surface Pro 7 was manufactured by Microsoft Corporation and purchased in Sydney, Australia. This two-part design was chosen for ease of integration with the RASS.

A white LED, a sheet of copper shim stock and a black “jiffy” box were used to create a point light source. A 0.3 mm diameter hole was created in the shim, which was secured to the box. The face of the LED was aligned with the hole and epoxied in place. A 50 Ω resistor in series limited the current to the device. This light source was connected to a BNC adaptor so it could easily be strobed with the Analog Discovery 2.

For these experiments, the light block was always mounted horizontally. The horizontal mounting means only vertical refractive index gradients can be imaged [16]. The horizontal orientation was chosen as the acoustic waves propagated upwards, meaning the resulting refractive index gradients were mostly vertical and could be imaged with a horizontal light block. It is shown later that a reasonable understanding of the turbulence can be obtained from variations in the vertical refractive index gradients. A sharp blade was used as the light block as it produced clearer images than the thin wire tested. A two-axis linear stage was used to adjust the height of the light block. The light block was positioned at the focus of the reflected light as per [16].

The Blackfly S camera, light block and light source were mounted to the wooden carrier board attached to a scissor jack so the height could be adjusted to align with the mirror, as shown in Figure 12. The board was then orientated such that the mirror was in view of the camera. The light block, light source and camera alignment were calibrated for each experiment. To calibrate the system, the wooden block was positioned such that the light block was at the focus of the light. The height of the light block was then adjusted so approximately 50% of the light was being blocked and the camera was producing a clear image.

The transducer array mounted below the mirror was turned on, and the acoustic waves were imaged. The acoustic waves were also imaged in the presence of turbulence generated by a heat gun on its coldest setting and directed through the beam.

### 3.3. Integrated System

To control the ambient light, a temporary darkroom was constructed in the available laboratory space.

The RASS was placed below the mirror such that acoustic waves propagated through the imaging region in front of it. For these tests, the RASS was placed in the near field of the acoustic array to maximize the illumination power density. The distance between the acoustic array and the mirror remained fixed throughout the experiments. Figure 13 shows the general configuration of the RASS and mirror used in the experiments.

As can be seen in Figure 14, the radar signal radiates from the horizontal green horn and is reflected upwards by a fine-wire grid mounted at 45° to the beam to pass through the imaging region in front of the mirror. The acoustic array (not visible) is mounted below the wire grid and passes through it unattenuated and thence through the mirror’s imaging region as well.

One image was taken for each acoustic burst as the camera was unable to support a higher frame rate. The ASUS laptop was used to interface with the Analog Discovery 2 using the provided SDK and was responsible for synchronizing the acoustic burst, LED burst and camera triggering. All control code was written in Python. Two separate computers were used as it was not possible to log data from the RASS sufficiently fast while also streaming images. When used in combination with the RASS, the LED was only strobed during the period when the acoustic signal was passing through the imaging region in front of the mirror. The camera took an image simultaneously. Images were given meaningful filenames and saved with time stamps. To ensure accurate synchronization, the 40 kHz acoustic signal output by the Analog Discovery 2 into the power amplifier was sampled by an analog input channel, where it triggered an accurate time stamp for post-processing. The time stamps and filenames were used to match each acoustic burst with the corresponding Schlieren image so they could be analyzed together.

Initially, a small but powerful fan was used to generate airflow that passed through a spiral vortex generator mounted within a white PVC pipe. Later, a leaf blower with a similar spiral vortex generator was used to generate more powerful turbulence. The turbulence from the leaf blower had a larger exit orifice and was significantly stronger, so it impacted a larger volume than the turbulence from the fan. In both cases, the vortex was directed above the acoustic transducer array and through the area imaged by the Schlieren imager, as shown in Figure 15.

Several scenarios with the turbulence generators were considered, and data was taken for each scenario.

The turbulence generator was on and directed towards the imaging region where Schlieren images are taken (above the transducer array);

The turbulence generator was turned on but pointed in a different direction;The turbulence generator was turned off.For all scenarios, the pipe containing the vortex generator remained in the same location, only the fan was rotated.

## 4. Results and Discussion

### 4.1. Schlieren Imager—Initial Results

For initial testing, the transducer array was removed from the RASS and placed directly below the mirror to image the emitted acoustic waves. The LED was strobed at the acoustic frequency to produce a standing wave. When the frequency of the acoustic wave was increased and the LED frequency remained constant, the acoustic wave appeared to move upwards. The reverse occurred when the acoustic wave was at a lower frequency than the strobe frequency. This is expected as outlined in [38,40,44], further verifying that the strobing was executed correctly. Images were taken both with and without a heat gun on its coldest setting acting as a disruption, as shown in Figure 16.

Note that the various airy discs and other spots that appear on the images are due to imperfections in the aged coating of the telescope mirror.

We were pleasantly surprised that this rather primitive setup was able to produce images of the acoustic pulse and that, as is clear from Figure 16, the acoustic waves were impacted by the turbulence from the heat gun. In this case, the acoustic waves were traveling upwards and the acoustic waves above the heat gun’s plume appear weaker than those that have not passed through it. Additionally, the acoustic waves directly in the heat gun’s plume appear slightly wider and more distorted than those that are not. These results confirmed that there was potential for detecting turbulent air with both RASS and Schlieren imaging, thus encouraging further investigation.

As the light block was oriented approximately horizontally, only vertical refractive index gradients could be visualized, as explained by [16,53]. The refractive index gradient produced by the acoustic wave varied vertically, so the horizontal light block placement offered the best method to visualize refractive index gradients. Although this method was able to detect turbulence, as is evident in Figure 16, by imaging the refractive index gradients along multiple dimensions we could obtain a greater understanding of the turbulence and its association with the RASS results. [30,54] propose light blocks that are capable of imaging refractive index gradients in two dimensions. Furthermore, implementing a quantitative technique such as BOS could allow for a greater understanding of the refractive index gradients [31,32,41]. However, these were not implemented as part of this investigation.

Unfortunately, the initial images of acoustic waves (Figure 16) are not evenly lit across the whole image, and this detracted somewhat from their effectiveness. However, as suggested in [16], additional care was taken during later experiments to ensure the light source and light block were better aligned to generate more uniformly illuminated images. The Schlieren imager was very sensitive to any variations in the placement of components, so re-calibration was required each time images were taken.

The sharp-blade light block used was satisfactory. Some initial testing was performed with a wire light block similar to [38,44,55], but the sharp blade produced clearer images. The superiority of the blade as a light block is unexpected given that the literature states a wire light block generally leads to fewer diffraction effects [30]. The blade used was very sharp, which minimized any diffraction effects [14].

The images show that the constructed Schlieren imager operated as intended and imaged small refractive index gradients as required.

### 4.2. RASS

The integrated system logged Doppler data from the RASS for offline processing. We used MATLAB to analyze the first 6 ms of received echo data after an acoustic burst was triggered. The start of each acoustic burst was robustly detected, and after 5.1 ms, the acoustic burst reached the roof of the darkroom, after which the results were invalid. However, there was a fixed 0.6 ms delay between triggering and the acoustic burst being output, so analyzing the first 6 ms after triggering a burst accounts for both of these factors.

Figure 17 shows the Doppler data from the RASS integrated over several acoustic bursts occurring within 10 s. The response for a single burst is shown for comparison. Both the envelope shape and sinusoidal nature of the signals match the previously accepted result in [48]; hence, recreating this measurement in MATLAB validates the performance-integrated system. It is believed that amplitude modulation of the RASS signal is caused primarily by reflections of the acoustic signal from the roof of the darkroom. However, as we are only interested in echoes from the first two ms of the return, these can be ignored.

### 4.3. Integrated System

The integrated system performed as expected, and Schlieren images were successfully taken while logging Doppler data from the RASS for all three scenarios outlined above. There is countless research detailing RASS and Schlieren imagers individually, but we were not able to find any work detailing a combined system, which limits the comparison between the observations in this discussion to the literature. However, the apparent gap in the literature emphasizes the novelty and innovation of our combined system.

The use of two computers to share the computational load enabled us to capture images and sample the Doppler signal at sufficiently high sampling rates for effective signal reconstruction and processing. However, this is rather cumbersome, so a more capable computer will be required in the future.

As mentioned in [16], there is a tradeoff between the contrast and the range of refractive index gradients that can be clearly imaged by a Schlieren imager. As the final system is optimized to capture images as the acoustic wave propagated through the imaging region of the Schlieren imager, the strobe duration is limited as are the number of pulses that can be integrated, hence the images produced were dark and with low contrast. This could be improved by increasing both the acoustic and radar power as well as the intensity of the light source. However, as this was impractical during the experimental period, image processing techniques were employed to improve the clarity of the images.

The image post-processing was completed in Python. Firstly, ten empty images (images with no acoustic wave or turbulence present) were integrated and subtracted from images with features to remove artifacts of the camera lens, the mirror and the pipe. Next, the mean from each image was subtracted. To ensure effective comparison between scenarios, the images were grouped into sets of three: one image with no turbulence, an image with the turbulence source turned on but pointed away, and an image with the turbulence source turned on and pointed towards the path of the RASS. Next, to contrast stretch the image, the 1st and 99th percentile of pixel intensities of the set of images were calculated in Python, and these values were mapped to the limits of the image’s dynamic range and the intermediate pixel values were linearly mapped across it. This effectively removed any significant outliers while avoiding excessive saturation. Images that were integrated or subtracted were all taken without recalibration in between to align the features across multiple images. A comparison of the images before and after post-processing is shown in Figure 18.

In some images, there is a bright artifact on the right side. Given the artifact is present when the turbulence generators were turned off and it is not the turbulence we are aiming to image, it is neglected when further analyzing the results. When this artifact was present, the images were cropped before post-processing so the artifact did not impact the resulting image.

The acoustic waves are clearly seen in all post-processed images. As expected, no turbulence is seen in the images where the fan is turned off. Significant turbulence is seen in the images where the turbulence source is turned on and directed towards the imaging area, but some turbulence is also seen in the images even when the turbulence source is pointed away from the imaging region of the Schlieren sensor. It is speculated that this is because there is always some turbulent air generated by the turbulence generator in the small, enclosed darkroom space.

The darkroom ensured the lighting was consistent and low throughout the experiments, ensuring the strobed LED was effective. Unfortunately, the geometry of the darkroom was not ideal, with the low roof limiting the maximum allowable range of the RASS and the generation of some standing waves. Though the short range was acceptable for these experiments, a larger range would be required to further analyze how variations in turbulence are reflected in a Doppler signal as it propagates through a larger volume of turbulence.

Similarly, the Schlieren imager was limited by the available mounting points for the components. The bench in the darkroom provided a stable place to mount components but it was not possible to adjust the height of the mirror, which limited the area that the Schlieren imager could visualize. In the future, a full-sized optical table in a larger dark room would be needed. Additionally, BOS could be used to image a large area by mounting a patterned background on the wall behind the imaging region and adjusting the height of the camera accordingly. BOS has previously been used on very large scales, which validates its use for increasing the observable area of experiments [55].

#### 4.3.1. Using the Fan to Generate Turbulence

We were able to capture Schlieren images and RASS Doppler data simultaneously. Acoustic bursts of ten cycles were used in this experiment as these optimize the tradeoff between resolution and intensity of the RASS return. Figure 19 shows Schlieren images and spectrograms with the small fan being used to generate turbulence. The spectrograms in the middle row along and the Schlieren images were each created over one acoustic burst. The Schlieren images and RASS data used in the spectrograms were taken simultaneously. The spectrograms in the bottom row show approximately 50 integrated acoustic bursts. Though there are visible differences in the Schlieren images in the three scenarios, it is clear that there are no discernible differences between the spectrograms of RASS signals in any of the scenarios considered.

The frequency domain signal in Figure 20 shows some small variations between the three scenarios. There is a peak at 1.2 kHz in the fan scenario that is not present when the fan is turned off or pointed away, which is very interesting. The main peak at the acoustic signal occurs at 43 kHz in the FFT and appears in all three scenarios with a similar magnitude. There are no clear differences between the scenarios.

Moreover, there are no discernible differences between the time-domain RASS signals in any of the scenarios considered. However, the time-domain signals are noisy and appear to include small phase differences, thus limiting quantitative analysis. It is possible that these small phase differences are a result of the acoustic bursts occurring at slightly different times, notwithstanding the careful synchronization maintained. These offsets were too small to be corrected in post-processing. Additionally, the amplitude of the time-domain signals appears to be approximately constant across the three scenarios.

When the fan was used to generate turbulence, there were clear differences between the three scenarios in the Schlieren images, but there were no significant differences in the RASS Doppler data. While the experiments with the fan show the integrated system can be used, stronger turbulence was needed to assess whether the RASS Doppler data could be impacted by turbulence.

#### 4.3.2. Using the Leaf blower to Generate Turbulence

Similar experiments were conducted with the leaf blower, and the results from both the Schlieren imaging and RASS Doppler data were promising. The experimental setup was not changed from the fan experiments other than the turbulence generation. Figure 21 shows Schlieren images and spectrograms taken for each of the three scenarios with the leaf blower. There is turbulence seen in the Schlieren images for both scenarios when the leaf blower is turned on. This suggests that although the leaf blower was pointed away from the RASS, there was still some airflow around the room resulting from the leaf blower being turned on. The acoustic waves are less clear in the image where the leaf blower is turned on and directed at the RASS, indicating that the turbulence is stronger. The Schlieren images in Figure 19 and Figure 21 show the Schlieren imager can consistently detect turbulence from different sources.

The spectrograms in Figure 21 show significant differences between the three scenarios for both a single burst and when many bursts are integrated. The RASS Doppler signal is weaker when the leaf blower is turned on, and there is a significant difference between when the leaf blower is directed away from the RASS and when it is directed towards the RASS. Figure A4 displays more examples of single-burst spectrograms and corresponding Schlieren images, showing consistent turbulence detection. The combined Schlieren images and spectrograms show that our system can detect turbulence simultaneously using a Schlieren imager and RASS.

Figure 22 and Figure 23 show the RASS Doppler signal in the time and frequency domain respectively. In both figures the turbulence appears to alter the RASS Doppler signal supporting the results observed in Figure 21. In the plots with several bursts integrated in Figure 22 we can see the amplitude of the RASS Doppler signal is slightly smaller when the leaf blower is turned on and directed towards the RASS. The RASS Doppler data from a single burst is noisy hence it is not possible to determine if the turbulence has an impact on the signal from a single burst. Unfortunately, both the single-burst and integrated signals again include a small phase difference limiting quantitative analysis.

In Figure 23, the main peak from the RASS Doppler signal is again observed at 43 kHz. For both a single burst and several bursts integrated the main peak has a smaller amplitude when the leaf blower is turned on and pointed towards the RASS. There are additional peaks at around 65 kHz. These peaks appear stronger when only a single burst is considered, indicating that they are not consistent. The 65 kHz peaks can be faintly seen in the single-burst spectrograms in Figure 21.

There is also a small peak for all three scenarios at below 2 kHz. There are two potential explanations. The first is that the leaf blower is introducing electrical noise. However, the more likely scenario is that the leaf blower is causing some microphonic effects in the imaging area, even when the airflow is directed away. This hypothesis is supported by the Schlieren images in Figure 21 showing some turbulence even when the leaf blower is turned on and pointed away. Further investigation is needed to confirm this hypothesis.

The turbulence generated by both the small fan and the leaf blower can be seen in the Schlieren images. However, the leaf blower has a much clearer impact on the Doppler signal than the small fan.

From these results, we conclude that the turbulence has an impact on the received Doppler signal. The impact of turbulence is observed in both single bursts and when several bursts are integrated. There is some research (e.g., [13,26,48]) that notes turbulence has an impact on the observed Doppler signal from an RASS, which agrees with our results.

Although we could not observe an impact of the turbulence in a single burst in the time domain, the observed changes in the spectrograms, frequency domain and corresponding Schlieren images are sufficient to conclude that turbulence can be detected simultaneously with an RASS and Schlieren imager.

## 5. Conclusions

We have achieved our goal of integrating RASS and Schlieren imaging to individually and simultaneously detect turbulence. To our knowledge, Schlieren imaging and RASS have not been combined in this manner before. Our integrated system provides a novel proof-of-concept that RASSs and Schlieren imaging systems can simultaneously observe turbulent atmospheric conditions. However, we have yet to provide a strong correlation between these measurement modalities.

Further investigation will be required to extend this system from simply detecting turbulence to its analysis. Any future work analyzing turbulence with RASS could continue to use Schlieren imaging to provide a ground truth. However, the integrated system would need to be redesigned to function over a larger range and more dimensions. BOS is a good candidate for analyzing refractive index gradients in multiple directions over large areas.

Future turbulence analysis could be combined with any relevant meteorological methods (e.g., [8,9,10]) for predicting turbulence. The ability to analyze turbulence would introduce potential use cases onboard small UAVs to implement pre-emptive control, which would allow UAVs to fly in more turbulent conditions [6,7]. The ultimate goal for this research is to dispense with the Schlieren component of the sensor and mount the RASS on a UAV. To achieve this, the current technology would need to be miniaturized by increasing the operational frequency of the Doppler radar to 94 GHz and ultrasound to 213 kHz, as shown in Figure 24. This is not a trivial undertaking due to the potential microphonic coupling between the acoustic and radar components of the sensor. However, as regards the Doppler radar component, one of the authors has more than thirty years of experience in developing millimeter wave radars, so that aspect of the development is not considered to be too challenging.

We successfully captured Doppler data with an RASS while simultaneously capturing Schlieren images. Turbulence was detected in both types of data captured. While there is still significant work required to apply RASSs for pre-emptive control on UAVs, we have provided a proof-of-concept that the technologies can be combined to detect turbulence.

## Figures and Tables

**Figure 1 sensors-23-08255-f001:**
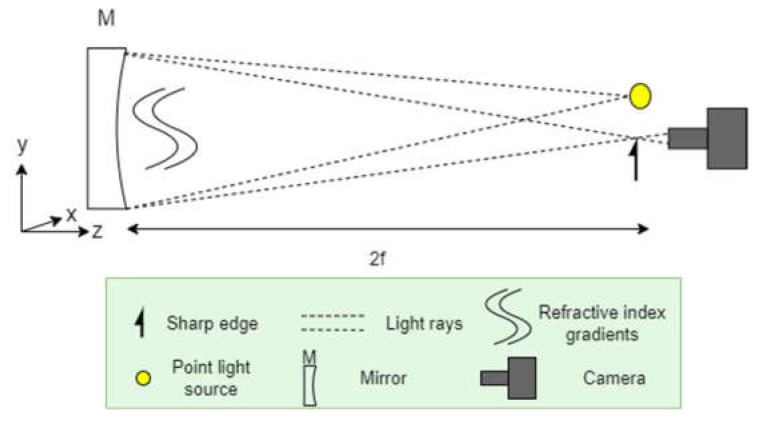
Schematic of a typical Schlieren imager with the mirror on the left and the light source, light block and camera on the right.

**Figure 2 sensors-23-08255-f002:**
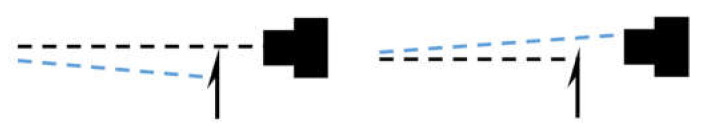
Schematic demonstrating the purpose of a light block with light rays that have not been perturbed (black) and light rays that have been refracted (blue). **Left**: light ray that would have passed the light block if not refracted is instead blocked. **Right**: light ray that would be blocked if not refracted instead passes.

**Figure 3 sensors-23-08255-f003:**
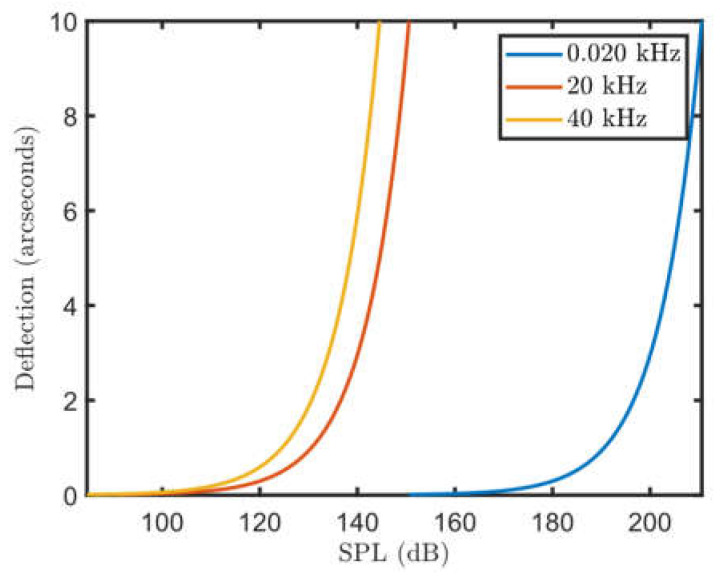
Relationship between deflection of electromagnetic wave, the SPL and the frequency of an acoustic wave being imaged with a Schlieren imager.

**Figure 4 sensors-23-08255-f004:**
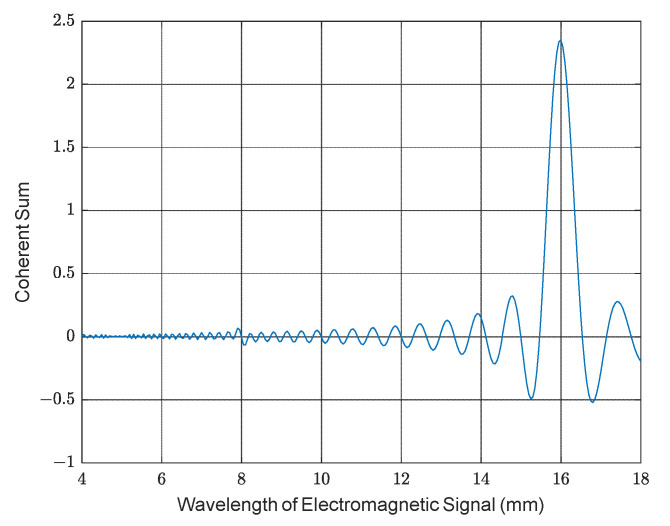
Simulation showing the coherent sum of the reflected signals as a function of wavelength to clearly illustrate the Bragg condition at an electromagnetic wavelength of 16 mm.

**Figure 5 sensors-23-08255-f005:**
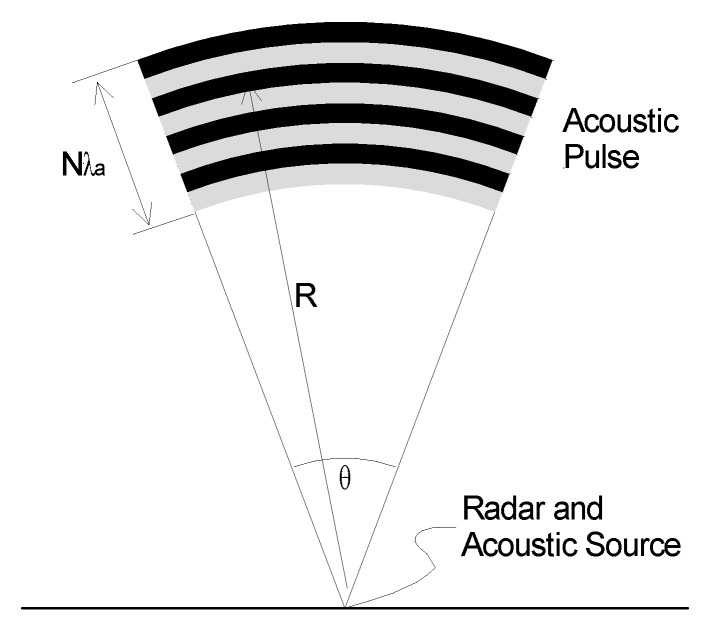
RASS geometry illustrating the effect of collocated sensors in which the reflected electromagnetic signal is focused back at the source to enhance the echo amplitude.

**Figure 6 sensors-23-08255-f006:**
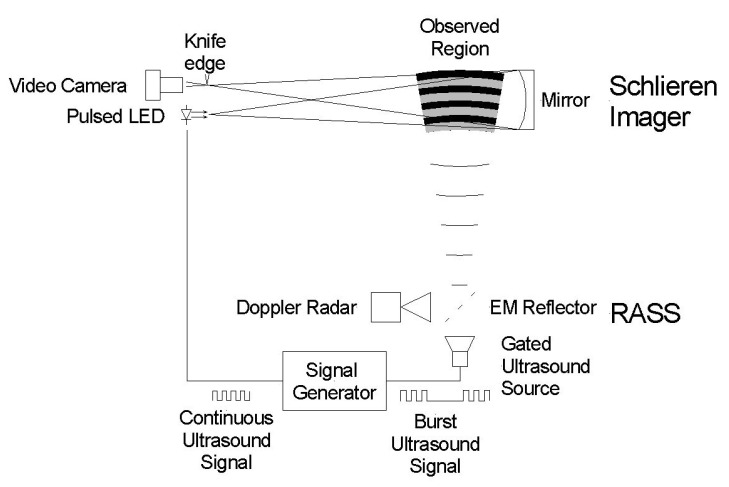
Schematic of the integrated system shows the important components of the RASS, including the acoustic and collocated Doppler radar as well as those of the Schlieren imager, consisting of the point LED source, mirror and camera. Synchronized ultrasonic signals drive the acoustic and optical components of the two sensors.

**Figure 7 sensors-23-08255-f007:**
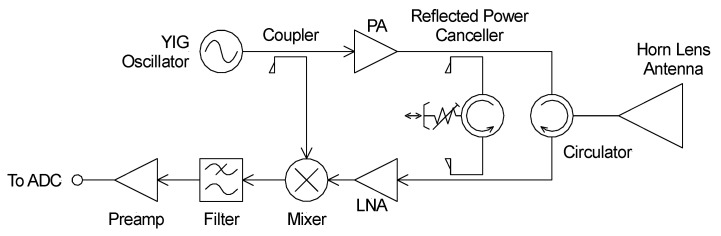
Schematic diagram of the Doppler radar and the incorporated reflected power canceller.

**Figure 8 sensors-23-08255-f008:**
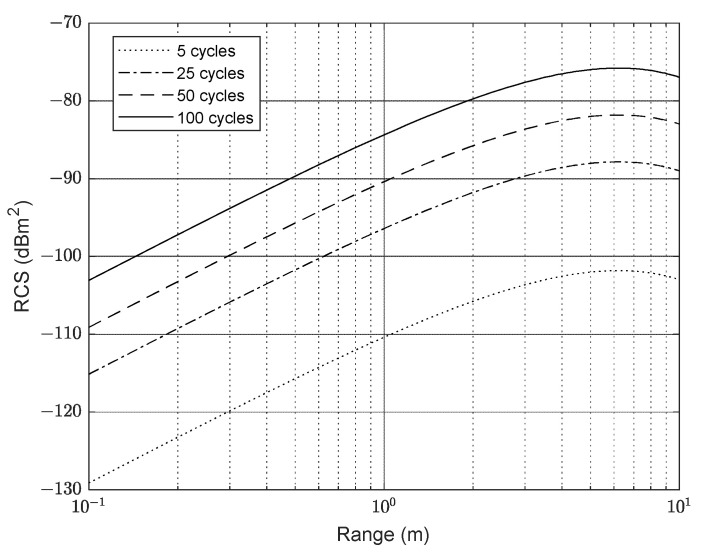
Radar cross-section of acoustic pulse with the number of cycles, N, in a pulse as a parameter.

**Figure 9 sensors-23-08255-f009:**
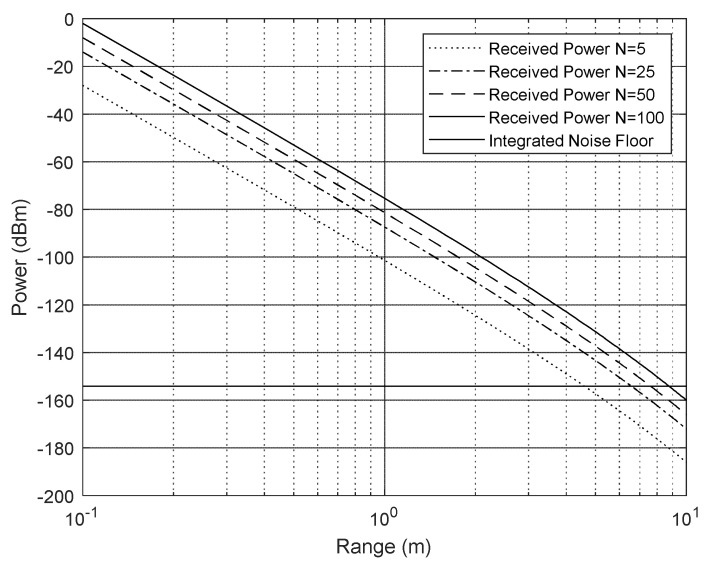
Received signal and noise levels with the number of cycles, N, in a pulse as a parameter.

**Figure 10 sensors-23-08255-f010:**
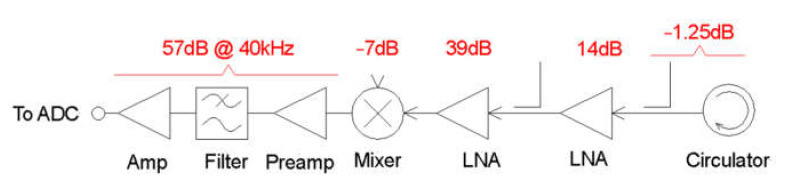
Block diagram showing the components of the receiver chain and their respective gains.

**Figure 11 sensors-23-08255-f011:**
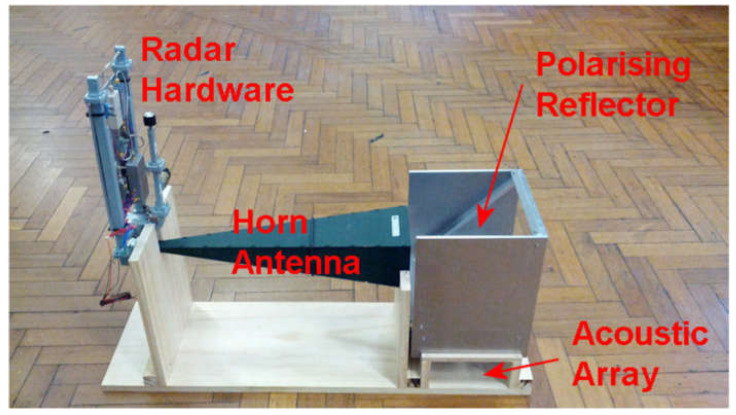
RASS used for turbulence experiments prior to integration with the Schlieren component of the system.

**Figure 12 sensors-23-08255-f012:**
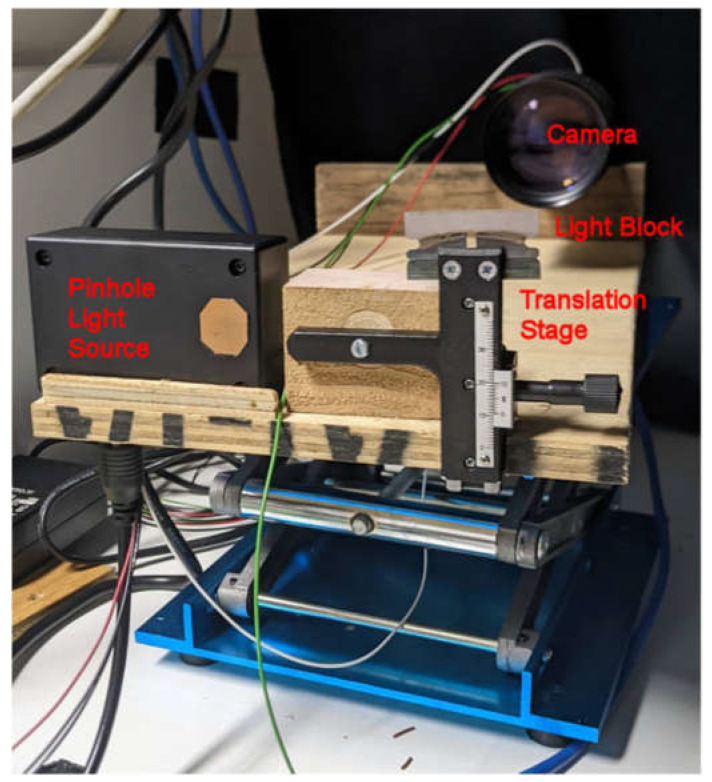
Light source, light block, and Blackfly S camera are mounted on a wooden board, with their height adjusted using a scissor jack.

**Figure 13 sensors-23-08255-f013:**
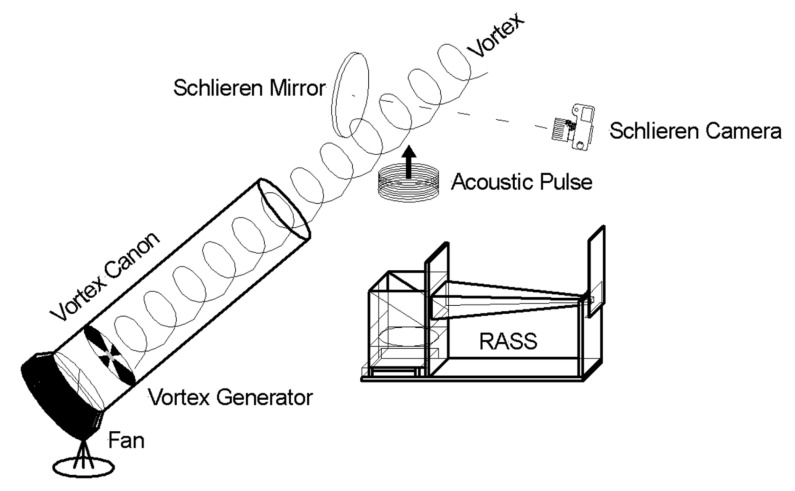
A schematic diagram showing the general configuration of the RASS and the Schlieren Imager.

**Figure 14 sensors-23-08255-f014:**
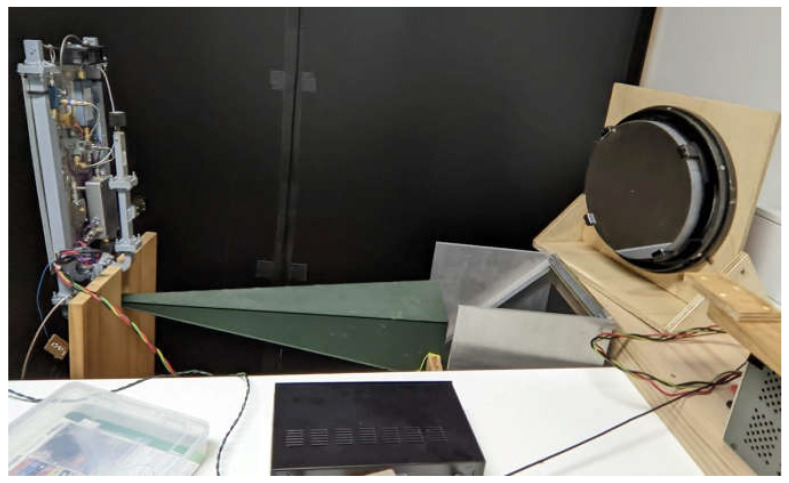
The radar signal for the RASS is radiated from the green horn and reflected upwards by a fine-wire grid mounted at 45° to the beam, while the acoustic signal is radiated through the grid and upwards in front of the parabolic mirror. The camera and light source are off the photo on the left and are pointed towards the mirror.

**Figure 15 sensors-23-08255-f015:**
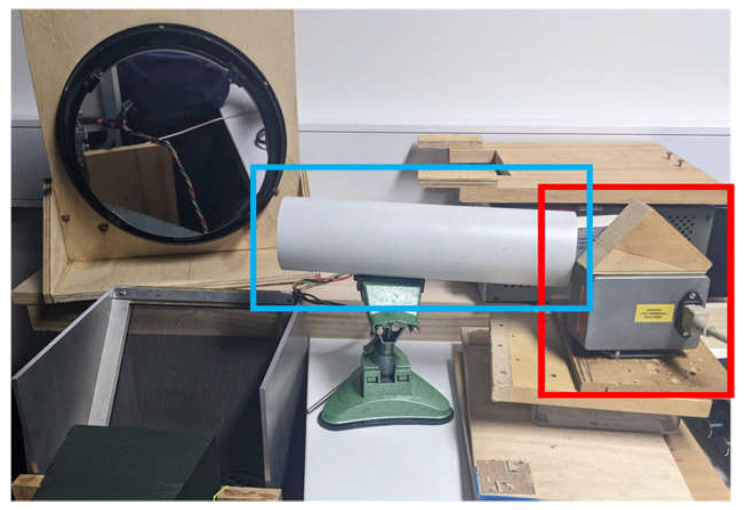
Fan placement in the darkroom. The fan is in the red box and the pipe with the internal vortex generator is in the blue box. To the left of the image are visible the mirror, the fine-wire grid and the end of the green horn.

**Figure 16 sensors-23-08255-f016:**
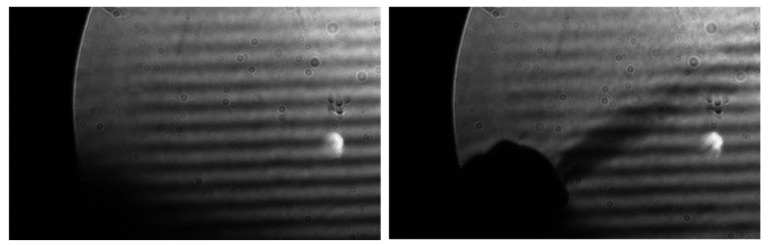
(**Left**): Acoustic waves from transducer imaged with Blackfly S camera. (**Right**): Acoustic waves with a heat gun on the coldest setting. The sound waves are traveling upwards. The acoustic waves above the heat gun’s plume are not as strong as the waves below.

**Figure 17 sensors-23-08255-f017:**
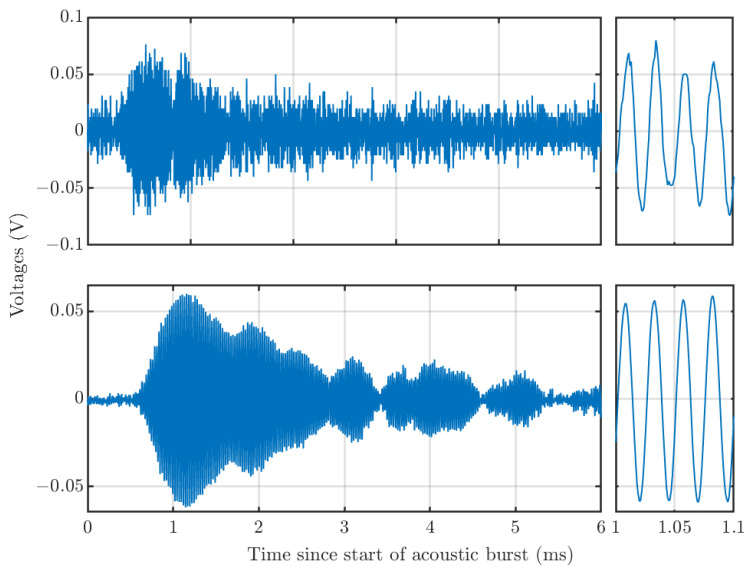
Top: EM signal received from RASS over the time period from the start of one acoustic burst to the next. Bottom: EM signal from RASS integrated over several acoustic bursts. The right panels show a short time span, displaying the sinusoidal nature of the received Doppler signals.

**Figure 18 sensors-23-08255-f018:**
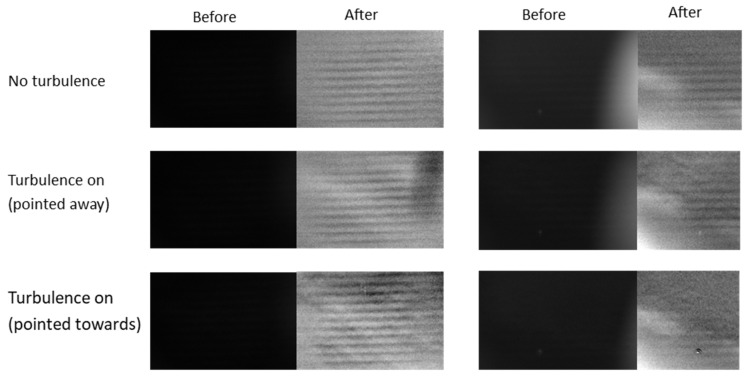
Schlieren images for all three scenarios before and after post-processing. **Left**: Fan used for turbulence generation. **Right**: Leaf blower used for turbulence generation. A bright artefact was seen on the right of some images, which was cropped out before post-processing.

**Figure 19 sensors-23-08255-f019:**
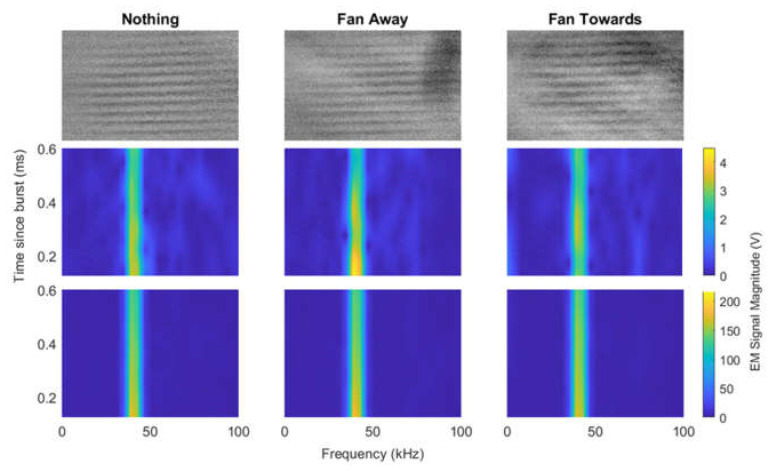
Schlieren images and spectrograms from experiments with the fan as a turbulence generator. The left column shows results with no turbulence. Middle column shows results with the fan turned on but pointed away. Right column shows result with the fan turned on and pointed towards the RASS. **Top**: Schlieren images of a single burst. **Middle**: Spectrograms of RASS Doppler data for a single burst. **Bottom**: Spectrograms of RASS Doppler data integrated over several bursts. While the turbulence can be seen in the Schlieren images, there is no significant difference between the scenarios in the spectrograms.

**Figure 20 sensors-23-08255-f020:**
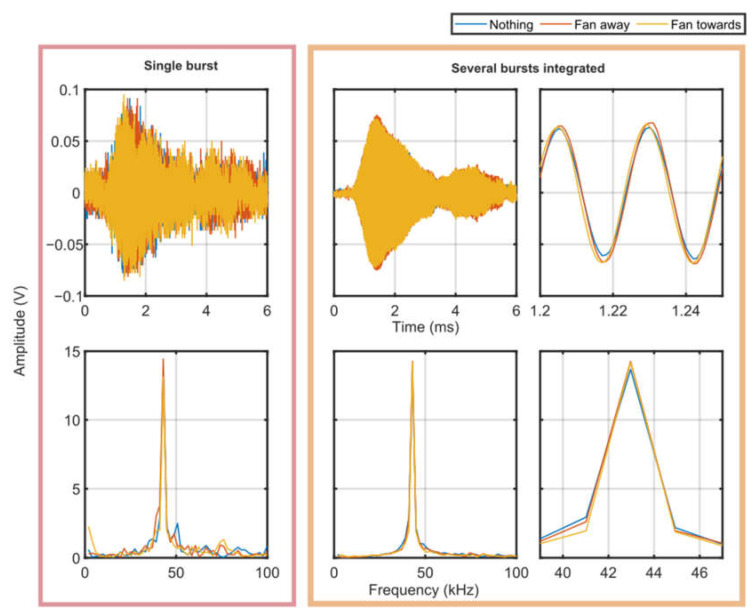
EM signal received from RASS during experiments using a fan to generate turbulence. The top plots show the time-domain response while the bottom row shows the EM signal in the frequency domain. The left plots are a single acoustic burst. The middle and right columns of plots show approximately 50 bursts integrated together. There are not any significant differences between the three scenarios with the fan turned off, the fan turned on and pointed away and the fan turned on and pointed in the path of the RASS.

**Figure 21 sensors-23-08255-f021:**
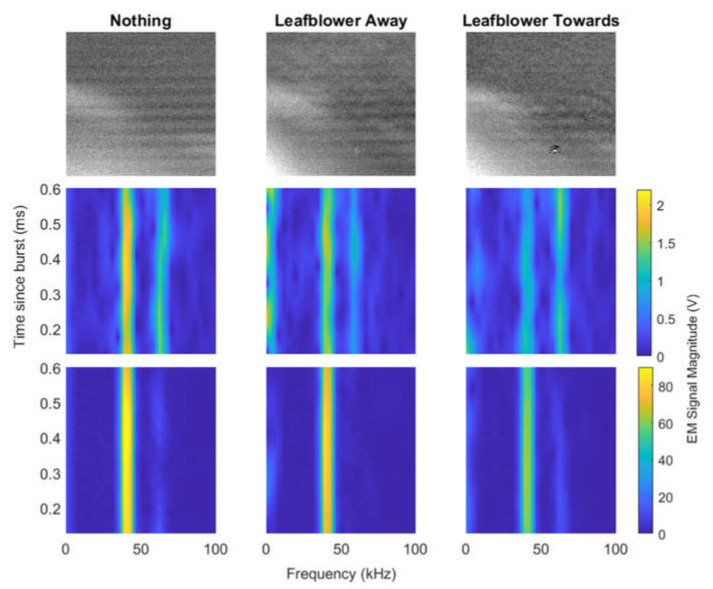
Schlieren images and spectrograms from experiments with leaf blower. The left column shows results with no turbulence. Middle column shows results with the leaf blower turned on but pointed away. Right column shows result with the leaf blower turned on and pointed towards the RASS. **Top**: Schlieren images of a single burst. **Middle**: Spectrograms of RASS Doppler data for a single burst. **Bottom**: Spectrograms of EM data from the RASS integrated over several bursts. In all of the rows, there is a clear difference between the three scenarios with the leaf blower.

**Figure 22 sensors-23-08255-f022:**
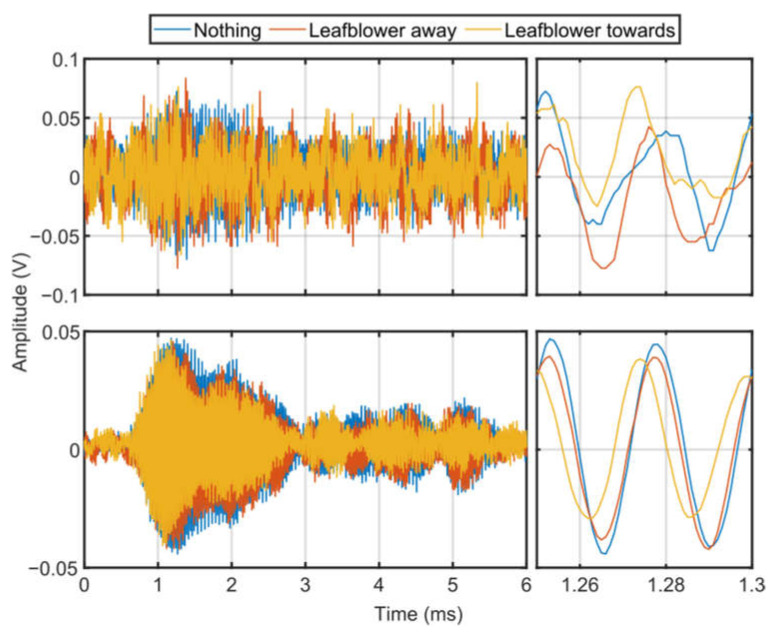
Time-domain plots of EM signal received from RASS during experiments using a leaf blower to generate turbulence. The top row shows a single burst, while the bottom row shows several bursts integrated. The left panels show the time-domain signal over the full time period considered, while the right panels show a smaller time period to emphasize the differences between the signals. The single-burst plots have too much noise to determine significant differences between the three scenarios. When the signals are integrated, the EM signal is weaker when the leaf blower is turned on and directed at the path of the RASS.

**Figure 23 sensors-23-08255-f023:**
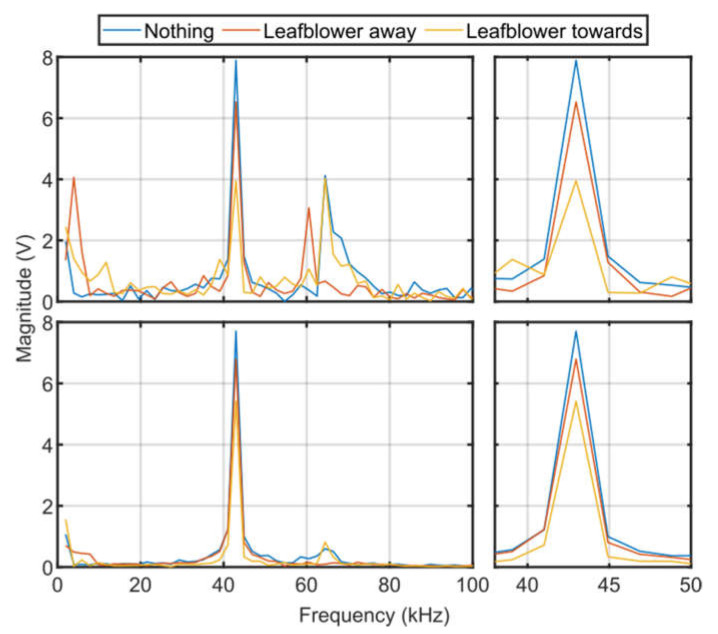
Frequency domain plots of EM signal received from RASS during experiments using a leaf blower to generate turbulence. The top row shows a single burst, while the bottom row shows several bursts integrated. The left panels show the time-domain signal over a 100 kHz bandwidth, while the right panels show a smaller frequency range to emphasize the differences between the signals. Both single-burst and integrated plots show differences between the three scenarios. The 43 kHz peak is strongest when the leaf blower is turned off and is weakest when the leaf blower is turned on a directed at the path of the RASS.

**Figure 24 sensors-23-08255-f024:**
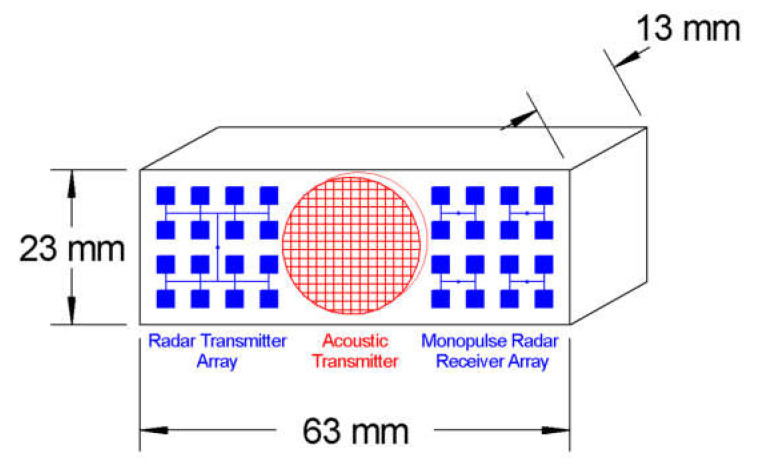
Sketch showing the proposed UAV based monopulse configuration.

## Data Availability

The data is available on request to the corresponding author.

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
