# Peer review of "Using Schlieren Imaging and a Radar Acoustic Sounding System for the Detection of Close-in Air Turbulenceâ€"

_sensors, 2023, doi:10.3390/s23198255_

Round 1
Reviewer 1 Report
I suggest that this manucript could be accepted by sensor.
Author Response
No comments to address.
Reviewer 2 Report
The review concerned an article entitled: Using Schlieren Imaging and Radar Acoustic Sounding System 2 for the Detection and Characterization of Close-in Air Turbulence.
In this paper, the authors introduce an innovative sensor designed to detect and characterize areas of air turbulence, with the aim of enhancing the stability of Unmanned Aerial Vehicles (UAVs) during adverse weather conditions. This novel sensor combines a Schlieren imager and a Radar Acoustic Sounding System (RASS) to generate dual-modality representations of air movement within the measurement volume.
Overall, this sensor represents an innovative approach to improving UAV stability in turbulent weather conditions. By combining two complementary imaging modalities, the authors aim to gain a deeper understanding of air turbulence and its impact on UAV performance, ultimately contributing to the development of more robust and weather-resistant UAV systems. In my opinion the article is worth consideration for publication.
Some questions and remarks to the text:
Introduction
1. What are the potential consequences of turbulence for aircraft, especially UAVs, as mentioned in the text?
2. What is the significance of the lidar-based sensor developed by the Japan Aerospace Exploration Agency (JAXA) in 2017 for detecting clear air turbulence (CAT)?
3. Why is it noteworthy that Schlieren imaging and RASS have been combined for detecting turbulence, as mentioned in the text?
4. When and how was the phenomenon of changes in refractive index induced by acoustic signals in air first identified?
5. What are some of the applications of RASS beyond examining air temperature, wind profiles, and turbulence in the lower troposphere mentioned in the text?
Operational Principles
1. Can you explain the difference between qualitative and quantitative Schlieren imaging systems, and what are their respective advantages?
2. What is Background Oriented Schlieren (BOS), and how does it enable quantitative measurement of refractive index gradients?
3. How can the fast propagation speed of acoustic waves be addressed when using Schlieren imaging, and what role does strobing the light source play in this context?
4. How can genuine standing waves be created when imaging ultrasonic acoustic waves with Schlieren systems, and why is this consideration important in the design of such imagers?
5. What is the recommended approach for accounting for atmospheric attenuation when calculating the effective RCS of acoustic signals?
Material and methods:
1. What were the specifications of the Doppler radar system used in this study, including its operational frequency, transmit power, antenna gain, and receive filter bandwidth?
2. How was the Doppler radar system calibrated and what role did the small Doppler target play in this calibration?
3. What was the reasoning behind the horizontal orientation of the Schlieren imager, and how was it calibrated for each experiment?
4. What methods were employed to generate turbulence for testing, and how were these scenarios controlled and recorded?
5. How were data collected and processed to ensure accurate synchronization and analysis of the acoustic and Schlieren images, particularly in the presence of turbulence?
Results and discussion:
1. What are the main components and setup used in the Schlieren Imager and RASS (Radio Acoustic Sounding System) experiments, and how do they work together to detect turbulence?
2. Why is the orientation of the light-block important in the Schlieren imaging process, and how does it relate to refractive index gradients and turbulence detection?
3. How did the experiments with the fan and leaf blower as turbulence sources differ in terms of turbulence generation and detection using both Schlieren imaging and RASS Doppler data?
4. What implications do these results have for the practical application of the integrated system in detecting and studying turbulence in different environments?
5. What are the limitations and potential improvements suggested for future experiments using the integrated Schlieren Imager and RASS system for turbulence detection?
Consculsions
1. How might increasing the operational frequency of the Doppler radar and ultrasound impact the miniaturization and performance of the technology for turbulence detection, and what challenges would need to be addressed in this process?
Reviewer 3 Report
The authors describe an experimental study wherein schlieren imaging and a radar acoustic sounding system (RASS) are simultaneously applied to detect and characterize air turbulence, which is proposed as a novel sensor for improving UAV stability.
The reviewer finds the experimental design to be generally sound and the manuscript to provide a clear explanation of the relevant background and experimental methods employed in the work. Nevertheless, several points need to be addressed before the manuscript should be considered for publication:
1. The abstract begins (lines 13-14) by describing the paper as presenting "...a novel sensor...it consists of combined schlieren imager and a RASS...". However, throughout the manuscript and in the conclusion, it becomes clear that only the RASS is being considered for use onboard a UAV, with the schlieren system being intended as providing a "ground truth reference" of the turbulence condition (lines 620-621). The descriptions of the combination of diagnostics used in this work and their intended uses needs to be made consistent throughout the manuscript.
2. Please ensure all acronyms are defined at their first usage (e.g., IFF in line 58)
3. The color of a purple box is referenced in the caption of Fig. 6, but the figure is grayscale in the manuscript.
4. Specific preliminary test cases for the schlieren imaging diagnostic are referenced (line 312), but no corresponding images are provided (e.g., in supplemental materials). If the author is going to include the detail that these configurations were tested, the reviewer would expect the reader to be provided with images by which they could visualize the resulting system performance.
5. Many of the non-idealities seen in the schlieren images (e.g., Airy discs (line 376), uneven illumination (line 401), etc.) could likely be addressed through the application of background normalization of the images using a bright-field image with no turbulence or acoustic forcing. This image processing step might improve the interpretability/effectiveness of the schlieren diagnostic.
6. In lines 425-427, the authors attribute the amplitude modulation of the RASS signal to the reflections of the EM signal from the roof of the dark room, which they say can be ignored because they are only interested in the first 2 ms of the signal. With EM radiation traveling at the speed of light (~1 ft/ns), reflections of the EM signal would be detected almost instantaneously. Please clarify this description an the reason for which the authors believe this mode of interference can be ignored.
7. In Fig. 21, the reason for a clear second peak in the spectrogram around 0 kHz appearing in the cases of no turbulence and "leafblower towards" but not the "leafblower away" case is not adequately discussed/explained.
Overall, the authors' concept of using schlieren imaging as a detailed reference for the performance of an RASS sensor is sound, and the particular feature of the ultrasonic acoustic waves being visible in the schlieren images is impressive. However, the impact of the paper is limited by the schlieren system not meeting its goal of providing a true "ground-truth" reference and instead being limited to vague, qualitative observations that add comparatively little to the interpretation of the RASS measurements.
Round 2
Reviewer 3 Report
Regarding the author's revisions to the manuscript in question in response to the previously raised points.
1. The authors state in their response that the manuscript was updated, but the abstract of the revised manuscript begins with the same description of the "novel sensor" that the reviewer raised as being problematic in the original manuscript. This must be addressed.
2. Okay
3. Additional description of Fig. 6 is warranted, either in the caption or through labeling the features described by the removed portion of the caption.
4. Okay
5. The image processing method of subtracting the mean of the image described in the paragraph beginning in line 495 is neither the same nor likely to be nearly as effective at correcting for spatial nonuniformity (e.g., the "bright artifact") as pixelwise normalization using a background image in which there is no turbulence or RASS signal. In the opinion of this reviewer, the manuscript would be made significantly more impactful by the authors' revisiting the image processing methodology such that the schlieren images could be made interpretable beyond observing the mere presence of turbulence.
The reviewer finds the characterization of their image processing as "fairly sophisticated" and producing "the best possible results under the circumstances" (lines 493-494) to be subjective, unsubstantiated, and likely inaccurate.
6. Okay
7. Okay
While the manuscript has been improved, the reviewer still finds the results presented to be insufficient to justify the conclusion that "We successfully detected turbulence with a RASS while simultaneously capturing Schlieren images to provide a ground truth of the turbulent conditions". While schlieren and RASS diagnostics were simultaneously employed, the lack of clarity in the schlieren images or application of any quantitative analysis precludes the schlieren results from being said to provide a "ground truth".
Okay, though the latest revisions may have introduced some issues
